# Analysis of the Sensing Margin of Silicon and Poly-Si 1T-DRAM

**DOI:** 10.3390/mi11020228

**Published:** 2020-02-23

**Authors:** Hyeonjeong Kim, Songyi Yoo, In-Man Kang, Seongjae Cho, Wookyung Sun, Hyungsoon Shin

**Affiliations:** 1Department of Electronic and Electrical Engineering, Ewha Womans University, Seoul 03760, Korea; laguswjd95@naver.com (H.K.); dbthddl0219@ewhain.net (S.Y.); 2School of Electronics Engineering, Kyungpook National University, Daegu 702–701, Korea; imkang@ee.knu.ac.kr; 3Department of Electronic Engineering, Gachon University, Gyeonggi-do 461–701, Korea; felixcho@gachon.ac.kr

**Keywords:** one-transistor dynamic random-access memory (1T-DRAM), polysilicon, grain boundary, electron trapping

## Abstract

Recently, one-transistor dynamic random-access memory (1T-DRAM) cells having a polysilicon body (poly-Si 1T-DRAM) have attracted attention as candidates to replace conventional one-transistor one-capacitor dynamic random-access memory (1T-1C DRAM). Poly-Si 1T-DRAM enables the cost-effective implementation of a silicon-on-insulator (SOI) structure and a three-dimensional (3D) stacked architecture for increasing integration density. However, studies on the transient characteristics of poly-Si 1T-DRAM are still lacking. In this paper, with TCAD simulation, we examine the differences between the memory mechanisms in poly-Si and silicon body 1T-DRAM. A silicon 1T-DRAM cell’s data state is determined by the number of holes stored in a floating body (FB), while a poly-Si 1T-DRAM cell’s state depends on the number of electrons trapped in its grain boundary (GB). This means that a poly-Si 1T-DRAM can perform memory operations by using GB as a storage region in thin body devices with a small FB area.

## 1. Introduction

Conventional one-transistor one-capacitor dynamic random-access memory (1T-1C DRAM) has reached its scaling limit due to the difficulty of miniaturizing capacitors. Therefore, capacitor-less one-transistor dynamic random-access memory (1T-DRAM), which does not need complicated capacitor fabrication, has been studied as a possible replacement for 1T-1C DRAM [1,2,3,4,5,6,7,8,9,10,11,12,13,14]. 1T-DRAM can be densely integrated because it has a small 4F^2^ cell size with a silicon-on-insulator (SOI) transistor as its basic structure. However, its memory performance deteriorates with decreased transistor size. As its channel length is decreased, its body needs to be thin enough to prevent short channel effects, thus reducing the floating body (FB) which is the charge storage region. In addition, 1T-DRAM is expensive because it requires the use of an SOI wafer. 

In recent years, poly-Si 1T-DRAM has gained much attention [15,16,17,18,19,20]. This type of device stores its charge in its grain boundary (GB) instead of in the FB, and therefore it is advantageous for short channel devices; its charge can be stored in the GB even with a thin body. In addition, an SOI structure with a poly-Si channel can be easily obtained by annealing deposited amorphous silicon. This feature facilitates a three-dimensional (3D) stack architecture that is cost effective and provides superior scalability. 

In this paper, we investigate the memory operating mechanisms of silicon 1T-DRAM and poly-Si 1T-DRAM by using device simulation. As a result, we reveal the reasons for the different memory operating characteristics of two 1T-DRAM by analyzing the changes in the devices’ charge and energy band diagrams. 

## 2. Simulation Method 

Simulations were performed to confirm the differences between 1T-DRAM operations according to their channel materials using the SENTAURUS TCAD tool. Figure 1a,b shows the cross-sections of the simulated devices. The structure is a single gate SOI transistor and the channel materials are single crystal silicon and polycrystalline silicon, respectively, as seen in the figure. The device parameters for the simulations were as follows: gate length (L_g_) = 200 nm, body thickness (T_body_) = 30/40/50 nm (typically 40 nm), gate oxide thickness (T_ox_) = 4 nm, buried oxide (T_box_) = 100 nm, source/drain doping concentration (N^+^: Arsenic) = 1 × 10^20^ cm^−3^, channel doping concentration (P: Boron) = 1 × 10^17^ cm^−3^, electron mobility (m_e_) = 200 cm^2^/V·s, and hole mobility (m_h_) = 100 cm^2^/V·s. Although both silicon and poly-Si body devices were simulated with the same mobility value, the poly-Si device reduced the drain current due to the charge trapping phenomenon in a single GB. It represented the mobility degradation of poly-Si as compared with silicon in the practical device. For the poly-Si channel, it was assumed that a single vertical grain boundary exists in the middle of the body to simplify simulation. The trap parameter was based on the work of [21]. In recent research, the energy band change as a function of the donor and acceptor type trap densities was confirmed; the investigators verified that donor type traps had little influence on band peak while acceptor type traps had a strong effect on it [18]. In addition, we studied the effect of the capture cross-section, (the probability of capturing carriers in the GB) on memory operating characteristics [22]. On the basis of this research, we chose a typical trap distribution for use in the simulation, as shown in the inset graph of Figure 1c [21,22,23]. The red and black lines, respectively, represent the donor type trap density and acceptor type trap density. The tails of the conduction and valance bands have an exponential distribution, and near the mid-gap, they have a Gaussian distribution. Figure 1c shows the transfer characteristics of 1T-DRAM with silicon and with poly-Si channels. For the poly-Si channel, the threshold voltage and the subthreshold swing values tend to increase as compared with the silicon channel because electrons are trapped in the GB under an inversion condition. Similarly, in the off state, the current of the poly-Si is smaller than that of the silicon, because less gate induced drain leakage (GIDL) occurs due to the reduction of the electric field in GB of poly-Si. 

Transient simulations were performed to investigate dynamic 1T-DRAM operations. Table 1 shows the operating bias conditions and time conditions for both the silicon and poly-Si devices. The write “1” operation provides hole charges to the device’s body using band-to-band tunneling, and the write “0” operation supplies electron charges to the body by applying a forward bias to the drain. In the read operation, a low drain voltage for non-destructive read and a gate voltage that maximized the sensing margin were applied. During the hold period, the gate and drain were set to 0 V to maintain the state of the cell. The write ”1” time and write ”0” time were set to a time sufficient for hole charge generation and electron charge supply, respectively. The hold time was set as a variable parameter to investigate retention characteristics of memory devices over time.

## 3. Results and Discussion

### 3.1. Transient Characteristics of Silicon and poly-Si 1T-DRAM

Figure 2a,b shows the transient characteristics of silicon and poly-Si 1T-DRAM cells for states “0” and “1” and three T_body_ values. The x-axis is the hold time, and the y-axis is the read current after writing “1” or “0”. With the sensing margin defined as the drain current difference (ΔI_DS_) between states “1” and “0” at the hold time of 10 ns, Table 2 shows the margin for the three T_body_ values for both devices. In both the silicon and poly-Si 1T-DRAMs, the sensing margin decreases in proportion to T_body._ However, the acceptable sensing margin is about 3 uA [4], and the sensing margin of silicon devices are smaller than 3uA regardless of T_body_. For the 50 nm and 30 nm T_body_ values, the sensing margin of poly-Si 1T-DRAM decreases only 30%, while that of silicon decreases by 90%, as shown Table 2. Another remarkable aspect shown in Figure 2a,b is that the read current increase/decrease trend is reversed between the two devices regardless of the body thickness. Silicon 1T-DRAM shows a decrease in state “1” current, while its state “0” current is maintained over time. In contrast, for the poly-Si 1T-DRAM, the read current of state “1” is maintained over time, and the read current of state “0” tends to increase. In order to analyze the cause of these opposing trends, the charges in the devices’ FB and GB at a 40 nm T_body_ are compared. 

Figure 3a shows the charge variation in the FB of silicon 1T-DRAM according to the hold time. The black lines represent the number of holes, and the red lines represent the number of electrons in each state. The figure shows that the number of electrons in the FB is similar for states “0” and “1”. However, when the hold time is small, the number of holes in state “1” is larger than it is in state “0” since the excess holes generated by the write “1” operation are stored in the FB. These holes lower the threshold voltage by forward biasing the FB. Therefore, the read current of state “1” is larger than that of state “0”. This difference disappears as the excess holes in state “1” decrease by recombination over time. In summary, the excess holes in state “1” play an important role in silicon 1T-DRAM state detection, as previously reported.

Figure 3b shows how the trapped charge varies with hold time in the GB of poly-Si 1T-DRAM. In contrast to silicon 1T-DRAM, the difference in the trapped hole charge between states “0” and “1” is small while the difference in the trapped electron charge is large. After a write “0” operation, the excess electrons are trapped in the GB, reducing the number of free electrons for the read operation. Therefore, the read current of state “0” is reduced for short hold times, as shown in Figure 2b. These trapped electrons detrap with time and the read “0” current gradually increases. This can be confirmed by observing the variation in the energy band diagram over time.

Figure 4 shows the conduction band energy diagrams of poly-Si 1T-DRAM at two hold times. The black and red lines are the diagrams for the hold times of 10^−8^ s and 10^−1^ s, respectively. After a write “0” operation, trapped electrons in the GB create a peak in the conduction band, as shown in Figure 4a. This energy peak is a barrier to current flow, therefore, the peak value of the conduction band is inversely proportional to the read current. Therefore, the poly-Si 1T-DRAM’s read “0” current is smaller than that of its read “1” current when the hold time is 10^−8^ s, as Figure 2b illustrates. However, the peak value decreases over time as trapped electrons in the GB detrap. This is consistent with the read “0” current increases with longer hold time, in Figure 2b. Unlike state “1”, a blue arrow in Figure 4a shows that there is large conduction band difference in state “0” for hold times of 10^−8^ s and 10^−1^ s. 

The result is that the read “1” current is constant up to 10^−1^ s as shown in Figure 2b. This indicates that poly-Si 1T-DRAM data states can be distinguished by the number of trapped electrons in state “0”.

We confirmed from our study results that the principle of data storage differs with the body material of 1T-DRAM. Silicon 1T-DRAM uses an FB as its hole storage region in the “1” state, and poly-Si 1T-DRAM uses a GB as an electron storage region in the “0” state. These results indicate why the sensing margin of silicon 1T-DRAM is dramatically reduced with decreasing T_body_, while the margin in poly-Si 1T-DRAM is little affected by T_body,_ as shown in Figure 2a,b. The reduction of FB space for holes in silicon 1T-DRAM means it cannot be scaled to short channel devices with its limited sensing margin for a thin T_body_. However, since poly-Si 1T-DRAM uses its GB for memory operation instead of an FB, it is suitable for short channel devices even in a thin body. 

### 3.2. The Effect of Grain Size on Sensing Margin Characteristics

A poly-Si 1T-DRAM’s GB is important to its characteristics. However, a random number of GBs are distributed in a channel since the grain size of poly-Si cannot be easily controlled. Therefore, a study of the memory characteristics’ dependence on the number of GBs in the channel and the grain size of poly-Si is necessary. In order to analyze this effect, structures having one to four GBs in poly-Si channel are used in our simulation, as shown in Figure 1b and Figure 5a–c. Since the grain size means the gap between GBs, many GBs in the channel indicate that the grain size is small in these figures. In addition, we examined the three structures with different GB locations and grain size for each number of GBs to investigate the effect of GB location or grain size. Figure 5d–f shows the simulated device structure with different locations for 3 GBs. In these figures, the grain size depends on the location. For example, the grain size of Figure 5d is 50 nm, whereas that of Figure 5f is 70 nm.

Figure 6 shows the sensing margin of poly-Si 1T-DRAM as a function of the number of GBs in the channel. In Figure 6a, the sensing margin is inversely proportional to the number of GBs. This is due to increasing trapped electron charge as the number of GBs increases, as shown in Figure 6b. On the one hand, more GBs mean that the trap density in the channel increases proportionally, and therefore trapped electron charge also increase. 

Since the trapped charge reduces free electrons in the poly-Si channel, the drain current of the device decreases, and this results in sensing margin degradation.

On the other hand, the GBs location and grain size have little influence on the sensing margin characteristics as compared with the number of GBs in the channel. Therefore, the number of GBs should be considered to predict the sensing margin of poly-Si 1T-DRAM.

## 4. Conclusions

This paper reports on work that a 1T-DRAM’s operating characteristics differ according to its body material. The paper also illustrates the causes of the differences by analyzing the change in charge and the energy band diagrams over time. In a conventional silicon 1T-DRAM, the excess holes in the FB produce a read current difference between the “0” and “1” state. In contrast, when the body material is polycrystalline silicon, it was confirmed that the number of trapped electrons in its GB plays an important role in state distinguishment. In addition, it was verified that the same characteristics are obtained even if the T_body_ is changed. 

A thin T_body_ in silicon 1T-DRAM significantly reduces the device’s sensing margin, while the poly-Si 1T-DRAM retains its margin. In terms of stability, it is important for large memory windows to remain stable for memory characteristics. However, due to the lack of storage region in a thin body device, silicon has a significantly small window. It means silicon 1T-DRAM cannot have stable memory characteristics. Therefore, poly-Si 1T-DRAM has an advantage for transistor scaling because it can operate in a thin body. In addition, it enables the fabrication of a 3D stacked structure that significantly improves the degree of integration. In order to further improve the performance of poly-Si 1T-DRAM, a study focused on how to efficiently store trapped electrons in the GB is required.

In addition, this paper confirms the effect of GB properties on the sensing margin characteristics of poly-Si 1T-DRAM. The sensing margin decreases in inverse proportion to the number of GBs in the polycrystalline silicon channel due to increased trapped electron charge. It was also verified that the number of GBs in the channel rather than their locations or grain size has a significant effect on the sensing margin characteristics of poly-Si 1T-DRAM.

## Figures and Tables

**Figure 1 micromachines-11-00228-f001:**
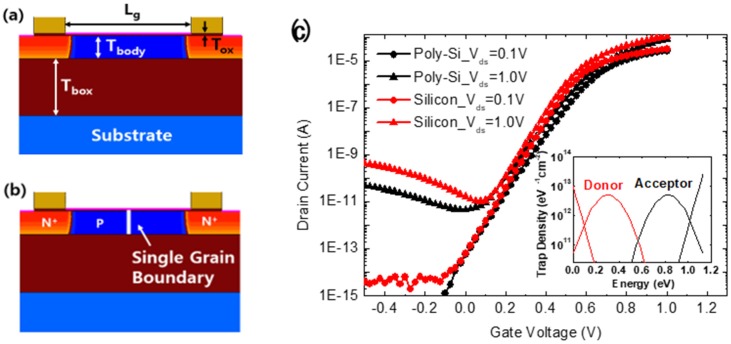
Cross-section of the simulated structure of (**a**) silicon one-transistor dynamic random-access memory (1T-DRAM); and (**b**) poly-Si 1T-DRAM; (**c**) transfer characteristics of silicon 1T-DRAM and poly-Si 1T-DRAM devices (inset, density of states used for poly-Si 1T-DRAM).

**Figure 2 micromachines-11-00228-f002:**
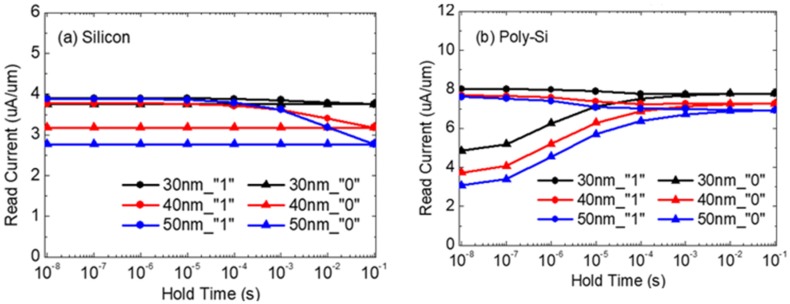
Transient characteristics of (**a**) silicon and (**b**) poly-Si 1T-DRAM according to T_body_/@T = 300 K.

**Figure 3 micromachines-11-00228-f003:**
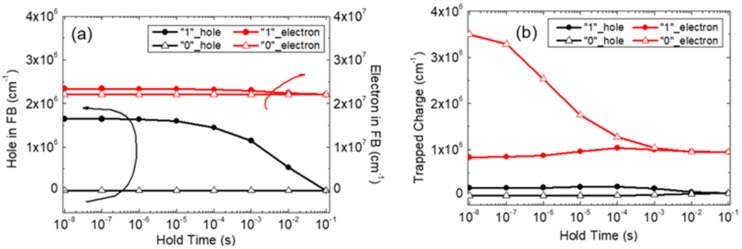
Charge variation according to the hold time (**a**) in the floating body (FB) of silicon 1T-DRAM and (**b**) in the grain boundary (GB) of poly-Si 1T-DRAM/@T = 300 K.

**Figure 4 micromachines-11-00228-f004:**
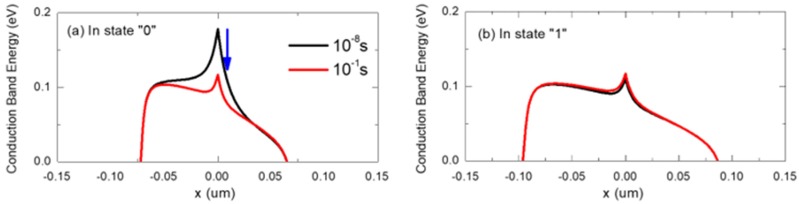
Conduction band energy diagram of poly-Si 1T-DRAM at two hold times (black lines, 10^−8^ s and red lines, 10^−1^ s). (**a**) During the read “0” period and (**b**) during the read “1” period.

**Figure 5 micromachines-11-00228-f005:**
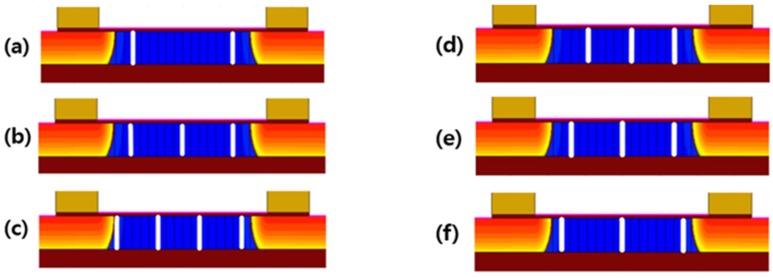
The simulated structure of poly-Si body 1T-DRAM with (**a**) 2 GBs; (**b**) 3 GBs; and (**c**) 4 GBs in a channel; (**d–f**) the simulated structure of 3 GBs poly-Si body 1T-DRAM with different GBs location /@T = 300 K.

**Figure 6 micromachines-11-00228-f006:**
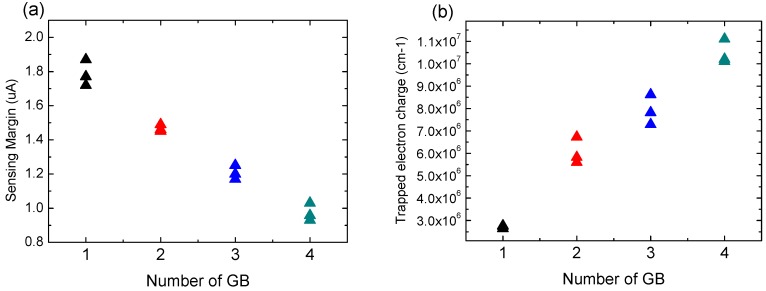
(**a**)Sensing margin and (**b**) trapped electron charge according to the number of GBs in the poly-Si channel of poly-Si 1T-DRAM.

**Table 1 micromachines-11-00228-t001:** Bias conditions for one-transistor dynamic random-access memory (1T-DRAM) operation.

	Write “1”	Write “0”	Read	Hold
**V_g_ (V)**	−2	0	0.6 (Silicon 1T-DRAM) 0.7 (Poly-Si 1T-DRAM) −	0
**V_d_ (V)**	2	−1.5	0.1	0
**Time (ns)**	500	150	10	−

**Table 2 micromachines-11-00228-t002:** Sensing margin of silicon and poly-Si 1T-DRAMs for varied T_body_.

T_body_	Sensing Margin of Silicon 1T- DRAM (uA/um)	Sensing Margin of Poly-Si 1T- DRAM (uA/um)
30 nm	0.15	3.16
40 nm	0.59	3.96
50 nm	1.11	4.54

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
