# Peer review of "Analysis of the Sensing Margin of Silicon and Poly-Si 1T-DRAM"

_micromachines, 2020, doi:10.3390/mi11020228_

Round 1

Reviewer 1 Report

The authors have revised the paper very well. The paper should be published in this journal.

Author Response

Thank you for your comments.

Reviewer 2 Report

Authors have satisfactorily respond to all the reviewer's comments. So, the paper can be now a good contribution to scientist community.

Author Response

Thank you for your comments.

This manuscript is a resubmission of an earlier submission. The following is a list of the peer review reports and author responses from that submission.

Round 1

Reviewer 1 Report

The authors have reported very good works on 1T-DRAM configuration concept. They have mentioned that poly-Si grain boundary can change the energy band as well as memory changes. This is novel and new work. This is also timely work for publication. The authors should explain the memory time or how long can store data in the revised manuscript. 

Reviewer 2 Report

In this manuscript the Authors present their investigations in capacitor less 1T-DRAM cells based on SOI substrates where the superficial silicon film has (a) mono-crystalline and (b) poly-crystalline nature. The investigations include only simulations. The overall quality of the paper is below because of the lack of scientific content. I would suggest the rejection of the manuscript in its present form and ask from the authors to resubmit it. Key points that must be improved are:

What are the poly-Si properties considered? The GB density plays a significant role in current flow and charge trapping, what is the GB density? what type are the traps considered at the grain boundaries? What is the role of GB in charge recombination? We need numbers in order to make a safe comparison. What is the carrier mobilities in poly-Si films? What are the values for acceptable memory window? What the acceptable values for charge retention? For the comparison between the c-Si and poly-Si based cells we need these figures. Indeed the memory window at the very initial time is large but decreases very rapidly. Is it good for the stable operation of a memory cell? What is the duration of the pulses in table 1? Are poly-Si cells as fast as c-Si? From fig.4 and 6 the c-Si has at least one decade better retention time than poly-Si. Why to consider poly-Si as channel material? The worst quality of poly-Si compared to c-Si affects also the p-n junctions between channel/S and D regions. How the quality of these contacts affects the poly-Si and c-Si caparcithorless memory cells?

Reviewer 3 Report

The paper is a comparison of two configuration of 1T-DRAM one with single crystalline silicon channel and the other with polysilicon body. 

The study is very simple since one only grain boundary is assumed. The grain size in polysilicon can not be easily controlled and different grain size can occur in a random way in devices like those described in this work. So, a study on the influence of number and size of the poysilicon grains should be performed in order to confirm the conclusions of this work.

On the other hand, authors found  (Figure 1) that the threshold voltage and the subthreshold swing values tend to increase compared to the silicon channel because electrons are trapped in the GB under the inversion  condition. But, there is another relevant difference: at negative voltage values, simulations predict that currents are much higher for the case of crystalline silicon. Typically, these currents are due to Gate Induced Barrier Lowering mechanisms. Authors should explain the reasons for these differences. 

On the other hand, current and charge transients  (retention curves) are only obtained during reading. In all cases the reading currents corresponding to "0" and "1" state trend to be equal for times longer than 0.1 s. That means that the memory must be read at very short times?. If that is the case, the  memory state is too much volatile, and this kind of technology would not have a practical interest. Authors should clarify this apparent weakness. 

Finally, one relevant aspect of any memory is the power  consumption during both writing and reading processes. Simulations of current transients during writing operation  and power consumption during both writing and reading must be include to a more complete comparation  of both technologies.

In summary, many improvements must be include in the work before being considered for publication.